# Community Detection on Evolving Graphs

**Aris Anagnostopoulos**
Sapienza University of Rome
aris@dis.uniroma1.it

**Jakub Łącki**
Sapienza University of Rome
j.lacki@mimuw.edu.pl

**Silvio Lattanzi**
Google
silviol@google.com

**Stefano Leonardi**
Sapienza University of Rome
leonardi@dis.uniroma1.it

**Mohammad Mahdian**
Google
mahdian@google.com

## Abstract

Clustering is a fundamental step in many information-retrieval and data-mining applications. Detecting clusters in graphs is also a key tool for finding the community structure in social and behavioral networks. In many of these applications, the input graph evolves over time in a continual and decentralized manner, and, to maintain a good clustering, the clustering algorithm needs to repeatedly probe the graph. Furthermore, there are often limitations on the frequency of such probes, either imposed explicitly by the online platform (e.g., in the case of crawling proprietary social networks like twitter) or implicitly because of resource limitations (e.g., in the case of crawling the web).

In this paper, we study a model of clustering on evolving graphs that captures this aspect of the problem. Our model is based on the classical stochastic block model, which has been used to assess rigorously the quality of various static clustering methods. In our model, the algorithm is supposed to reconstruct the planted clustering, given the ability to query for small pieces of local information about the graph, at a limited rate. We design and analyze clustering algorithms that work in this model, and show asymptotically tight upper and lower bounds on their accuracy. Finally, we perform simulations, which demonstrate that our main asymptotic results hold true also in practice.

## 1 Introduction

This work studies the problem of detecting the community structure of a dynamic network according to the framework of *evolving graphs* [3]. In this model the underlying graph evolves over time, subject to a probabilistic process that modifies the vertices and the edges of the graph. The algorithm can learn the changes that take place in the network only by probing the graph at a limited rate. The main question for the evolving graph model is to design strategies for probing the graph, such as to obtain information that is sufficient to maintain a solution that is competitive with a solution that can be computed if the entire underlying graph is known.

The motivation for studying this model comes from the the inadequacy of the classical computational paradigm, which assumes perfect knowledge of the input data and an algorithm that terminates. The evolving graph model captures the evolving and decentralized nature of large-scale online social networks. An important part of the model is that only a limited number of probes can be made at each time step. This assumption is motivated by the limitations imposed by many social network platforms such as Twitter or Facebook, where the network is constantly evolving and the access to the structure is possible through an API that implements a rate-limited oracle. Even in cases where such rate-limits are not exogenously imposed (e.g., when the network under consideration is the Web),

resource constraints often prohibit us from making too many probes in each time step (probing a large graphs stored across many machines is a costly operation). The evolving graph model has been considered for PageRank computation [4] and connectivity problems [3]. This work is the first to address the problem of community detection in the evolving graph model.

Our probabilistic model of the evolution of the community structure of a network is based on the *stochastic block model* (SBM) [1, 2, 5, 10]. It is a widely accepted model of probabilistic networks for the study of community-detection methods, which generates graphs with an embodied community structure. In the basic form of the model, vertices of a graph are first partitioned into $k$ disjoint communities in a probabilistic manner. Then, two nodes of the same community are linked with probability $p$, and two nodes of distinct communities are linked with probability $q$, where $p > q$. All the connections are mutually independent.

We make a first step in the study of community detection in the evolving-graph model by considering an *evolving stochastic block model*, which allows nodes to change their communities according to a given stochastic process.

## 1.1 Our Contributions

Our first step is to define a meaningful model for community detection on evolving graphs. We do this by extending the stochastic block model to the evolving setting. The *evolving stochastic block model* generates an $n$-node graph, whose nodes are partitioned into $k$ communities. At each time step, some nodes may change their communities in a random fashion. Namely, with probability $1/n$ each node is reassigned; when this happens it is moved to the $i$th community $C_i$ (which we also call a *cluster $C_i$*) with probability $\alpha_i$, where $\{\alpha_i\}_{i=1}^k$ form a probability distribution. After being reassigned, the neighborhood of the node is updated accordingly.

While these changes are being performed, we are unaware of them. Yet, at each step, we have a budget of $\beta$ queries that we can perform to the graph (later we will specify values for $\beta$ that allow us to obtain meaningful results—a value of $\beta$ that is too small may not allow the algorithm to catch up with the changes; a value that is too large makes the problem trivial and unrealistic). A query of the algorithm consists in choosing a single node. The result of the query is the list of neighbors of the chosen node at the moment of the query. Our goal is to design an algorithm that is able to issue queries over time in such a way that at each step it may report a partitioning $\{\hat{C}_1, \ldots, \hat{C}_k\}$ that is as close as possible to the real one $\{C_1, \ldots, C_k\}$. A difficulty of the evolving-graph model is that, because we observe the process for an infinite amount of time, even events with negligible probability will take place. Thus we should design algorithms that are able to provide guarantees for most of the time and recover even after highly unlikely events take place.

Let us now present our results at a high level. For simplicity of the description, let us assume that $p = 1$, $q = 0$ and that the query model is slightly different, namely the algorithm can discover the entire contents of a given cluster with one query.[1]

We first study algorithms that at each step pick the cluster to query independently at random from some predefined distribution. One natural idea is to pick a cluster proportionally to its size (which is essentially the same as querying the cluster of a node chosen uniformly at random). However, we show that a better strategy is to query a cluster proportionally to the *square root* of its size. While the two strategies are equivalent if the cluster probabilities $\{\alpha_i\}_{i=1}^k$ are uniform, the latter becomes better in the case of skewed distributions. For example, if we have $n^{1/3}$ clusters, and the associated probabilities are $\alpha_i \sim 1/i^2$, the first strategy incorrectly classifies $O(n^{1/3})$ nodes in each step (in expectation), compared to only $O(\log^2 n)$ nodes misclassified by the second strategy. Furthermore, our experimental analysis suggests that the the strategy of probing a cluster with a frequency proportional to the square root of its size is not only efficient in theory, but it may be a good choice for practical application as well.

We later improve this result and give an algorithm that uses a mixture of cluster and node queries. In the considered example when $\alpha_i \sim 1/i^2$, at each step it reports clusterings with only $O(1)$ misclassified nodes (in expectation). Although the query strategy and the error bound expressed in

terms of $\{\alpha_i\}_{i=1}^k$ are both quite complex, we are able show that the algorithm is optimal, by giving a matching lower bound.

Finally, we also show how to deal with the case when $1 \geq p > q \geq 0$. In this case querying node $v$ provides us with only partial information about its cluster: it is connected to only a subset of the nodes in $C$. In this case we impose some assumptions on $p$ and $q$, and we provide an algorithm that given a node can discover the entire contents of its cluster with $O(\log n/p)$ node queries. This algorithm allows us to extend the previous results to the case when $p > q > 0$ (and $p$ and $q$ are sufficiently far from each other), at the cost of performing $\beta = O(\log n/p)$ queries per step. Even though the evolving graph model requires the algorithm to issue a low number of queries, our analysis shows that (under reasonable assumptions on $p$ and $q$) this small number of queries is sufficient to maintain a high-quality clustering.

Our theoretical results hold for large enough $n$. Therefore, we also perform simulations, which demonstrate that our final theoretically optimal algorithm is able to beat the other algorithms even for small values of $n$.

## 2   Related Work

Clustering and community-detection techniques have been studied by hundreds of researchers. In social networks, detecting the clustering structure is a basic primitive for finding communities of users, that is, sets of users sharing similar interests or affiliations [12, 16]. In recommendation networks cluster discovery is often used to improve the quality of recommendation systems [13]. Other relevant applications of clustering can be found in image processing, bioinformatics, image analysis and text classification.

Prior to the evolving model, a number of dynamic computation models have been studied, such as online computation (the input data are revealed step by step), dynamic algorithms and data structures (the input data are modified dynamically), and streaming computation (the input data are revealed step by step while the algorithm is space constrained). Hartamann et al. [9] presented a survey of results for clustering dynamic networks in some of the previously mentioned models. However, none of the aforementioned models capture the relevant features of the dynamic evolution of large-scale data sets: the data evolves at a slow pace and an algorithm can learn the data changes only by probing specific portions of the graph at some cost.

The stochastic block model, used by sociologists [10], has recently received a growing attention in computer science, machine learning, and statistics [1, 2, 5, 6, 17]. At the theoretical level, most work has studied the range of parameters, for which the communities can be recovered from the generated graph, both in the case of two [1, 7, 11, 14, 15] or more [2, 5] communities.

Another line of research focused on studying different dynamic versions of the stochastic block model [8, 18, 19, 20]. Yet, there is a lack of theoretical work on modeling and analyzing stochastic block models, and more generally community detection on evolving graph. This paper makes the first step in this direction.

## 3   Model

In this paper we analyze an evolving extension of the stochastic block model [10]. We call this new model the *evolving stochastic block model*. In this model we consider a graph of $n$ nodes, which are assigned to one of $k$ clusters, and the probability that two nodes have an edge between them depends on the clusters to which they are assigned. More formally, consider a probability distribution $\alpha_1, \ldots, \alpha_k$ (i.e., $\alpha_i > 0$ and $\sum_i \alpha_i = 1$). Without loss of generality, throughout the paper we assume $\alpha_1 \geq \ldots \geq \alpha_k$. Also, for each $1 \leq i \leq k$ we also assume that $\alpha_i < 1 - \epsilon_\alpha$ for some constant $0 < \epsilon_\alpha < 1$.

At the beginning, each node independently picks one of the $k$ clusters. The probability that the node picks cluster $i$ is $\alpha_i$. We denote this clustering of the nodes by $\mathcal{C}$. Nodes that pick the same cluster $i$ are connected with a fixed probability $p_i$ (which may depend on $n$), whereas pairs of nodes that pick two different clusters $i$ and $j$ are connected with probability $q_{ij}$ (also possibly dependent on $n$). Note that $q_{ij} = q_{ji}$ and the edges are independent of each other. We denote $p := \min_{1 \leq i \leq k} p_i$ and $q := \max_{1 \leq i,j \leq k} q_{i,j}$.

So far, our model is very similar to the classic stochastic block model. Now we introduce its main distinctive property, namely the evolution dynamics.

**Evolution model:** In our analysis, we assume that the graph evolves in discrete time steps indexed by natural numbers. The nodes change their cluster in a random manner. At each time step, every node $v$ is *reassigned* with probability $1/n$ (independently from other nodes). When this happens, $v$ first deletes all the edges to its neighbors, then selects a new cluster $i$ with probability $\alpha_i$ and finally adds new edges with probability $p_i$ to nodes in cluster $i$ and with probability $q_{ij}$ to nodes in cluster $j$, for every $j \neq i$. For $1 \leq i \leq k$ and $t \in \mathbb{N}$, we denote by $C_i^t$ the set of nodes assigned to cluster $i$ just after the reassignments in time step $t$. Note that we use $C_i$ to denote the cluster itself, but $C_i^t$ to denote its contents.

**Query model:** We assume that the algorithm may gather information about the clusters by issuing queries. In a single query the algorithm chooses a single node $v$ and learns the list of current neighbors of $v$. In each time step, the graph is probed after all reassignments are made.

We study algorithms that learn the cluster structure of the graph. The goal of our algorithm is to report an approximate clustering $\hat{\mathcal{C}}$ of the graph *at the end of each time step*, that is close to the true clustering $\mathcal{C}$. We define the distance between two clusterings (partitions) $\mathcal{C} = \{C_1, C_2, \ldots, C_k\}$ and $\hat{\mathcal{C}} = \{\hat{C}_1, \hat{C}_2, \ldots, \hat{C}_k\}$ of the nodes as

$$d(\mathcal{C}, \hat{\mathcal{C}}) = \min_{\pi} \sum_{i=1}^{k} |C_i \triangle \hat{C}_{\pi(i)}|,$$

where the minimum is taken over all the permutations $\pi$ of $\{1, \ldots, k\}$, and $\triangle$ denotes the symmetric difference between two sets, i.e., $A \triangle B = (A \setminus B) \cup (B \setminus A)$.[2] The distance $d(\mathcal{C}, \hat{\mathcal{C}})$ is called the *error* of the algorithm (or of the returned clustering).

Finally, in our analysis we assume that $p$ and $q$ are far apart, more formally we assume that:

**Assumption 1.** *For every $i \in [k]$, and parameters $K$, $\lambda$ and $\lambda'$ that we fix later, we have: (i) $p\alpha_i > Kq$, (ii) $p^2\alpha_i n \geq \lambda \log n$ and (iii) $p\alpha_i n \geq \lambda' \log n$.*

Let us now discuss the above assumptions. Observe that (iii) follows from (ii). However, we prefer to make them separate, as we mostly rely only on (iii). Assumption (iii) is necessary to assure that most of the nodes in the cluster have at least a single edge to another node in the same cluster. In the analysis, we set $\lambda'$ to be large enough (yet, still constant), to assure that for every given time $t$ each node has $\Omega(\log n)$ edges to nodes of the same cluster, with high probability.

We use Assumption 1 in an algorithm that, given a node $v$, finds all nodes of the cluster of $v$ (correctly with high probability[3]) and issues only $O(\log n/p)$ queries. Our algorithm also uses (ii), which is slightly stronger than (iii) (it implies that two nodes from the same cluster have many neighbors in common), as well as (i), which guarantees that (on average) most neighbors of a node $v$ belong to the cluster of $v$.

**Discussion:** The assumed graph model is relatively simple—certainly not complex enough to claim that it accurately models real-world graphs. Nevertheless, this work is the first attempt to formally study clustering in dynamic graphs and several simplifying assumptions are necessary to obtain provable guarantees. Even with this basic model, the analysis is rather involved. Dealing with difficult features of a more advanced model would overshadow our main findings.

We believe that if we want to keep the number of queries low, that is, $O(\log n/p)$, Assumption 1 cannot be relaxed considerably, that is, $p$ and $q$ cannot be too close to each other. At the same time, recovery of clusters in the (nonevolving) stochastic block model has also been studied for stricter ranges of parameters. However, the known algorithms in such settings inspect considerably more nodes and require that the cluster probabilities $\{\alpha_i\}_{i=1}^{k}$ are close to being uniform [5]. The results that apply to the case with many clusters with nonuniform sizes require that $p$ and $q$ are relatively far apart. We note that in studying the classic stochastic block model it is a standard assumption to know $p$ and $q$, so we also assume it in this work for the sake of simplicity.

Our model assumes that (in expectation) only one node changes its cluster at every time step. However, we believe that the analysis can be extended to the case when $c > 1$ nodes change their cluster every step (in expectation) at the cost of using $c$ times more queries.

Generalizing the results of this paper to more general models is a challenging open problem. Some interesting directions are, for example, using graphs models with overlapping communities or analyzing a more general model of moving nodes between clusters.

## 4 Algorithms and Main Results

In this section we outline our main results. For simplicity, we omit some technical details, mostly concerning probability. In particular, we say that an event happens with high probability, if it happens with probability at least $1 - 1/n^c$, for some constant $c > 1$, but in this section we do not specify how this constant is defined.[4]

We are interested in studying the behavior of the algorithm in an arbitrary time step. We start by stating a lemma showing that to obtain an algorithm that can run indefinitely long, it suffices to designing an algorithm that uses $\beta$ queries per step, initializes in $O(n \log n)$ steps and works with high probability for $n^2$ steps.

**Lemma 1.** *Assume that there exists an algorithm for clustering evolving graphs that issues $\beta$ queries per step and that at each time step $t$ such that $t = \Omega(n \log n)$ and $t \leq n^2$ it reports a clustering of expected error $E$ correctly with high probability.*

*Then, there exists an algorithm for clustering evolving graphs that issues $2\beta$ queries per step and at each time step $t$ such that $t = \Omega(n \log n)$ it reports a clustering of expected error $O(E)$.*

To prove this lemma, we show that it suffices to run a new instance of the assumed algorithm every $n^2$ steps. In this way, when the first instance is no longer guaranteed to work, the second one has finished initializing and can be used to report clusterings.

### 4.1 Simulating Node Queries

We now show how to reduce the problem to the setting in which an algorithm can query for the entire contents of a cluster. This is done in two steps. As a first step, we give an algorithm for detecting the cluster of a given node $v$ by using only $O(\log n/p)$ node queries.

This algorithm maintains *score* of each node in the graph. Initially, the scores are all equal to $0$. The algorithm queries $O(\log n/p)$ neighbors of $v$ and adds a score of $1$ to every neighbor of neighbor of $v$. We use Assumption 1 to prove that after this step, with high probability there is a gap between the minimum score of a node inside the cluster of $v$ and the maximum score of a node outside it.

**Lemma 2.** *Suppose that Assumption 1 holds. Then, there exists an algorithm that, given a node $v$, correctly identifies all nodes in the cluster of $v$ with high probability. It issues $O(\log n/p)$ queries.*

Observe that Lemma 2 effectively reduces our problem to the case when $p = 1$ and $q = 0$: a single execution of the algorithm gives us the entire cluster of a node, just like a single query for this node in the case when $p = 1$ and $q = 0$.

In the second step, we give a data structure that maintains an approximate clustering of the nodes and detects the number of cluster $k$ together with (approximate) cluster probabilities. Internally, it uses the algorithm of Lemma 2.

**Lemma 3.** *Suppose that Assumption 1 holds. Then there exists a data structure that at each time step $t = \Omega(n)$ may answer the following queries:*

1. *Given a cluster number $i$, return a node $v$, such that $\mathbf{Pr}(v \in C_i^t) \geq 1/2$.*

2. *Given $C_i^t$ (the contents of cluster $C_i$) return $i$.*

3. *Return $k$ and a sequence $\alpha_1', \ldots, \alpha_k'$, such that for each $1 \leq i \leq k$, we have $\alpha_i/2 \leq \alpha_i' \leq 3\alpha_i/2$.*

*The data structure runs correctly for $n^2$ steps with high probability and issues $O(\log n/p)$ queries per step.*

*Furthermore if $p = 1$ and $q = 0$, it makes only 1 query per step.*

Note that because the data structure can only use node queries to access the graph, it imposes its own numbering on the clusters that it uses consistently. Let us now describe the high-level idea behind it.

In each step the data structure selects a node uniformly at random and discovers its entire cluster using the algorithm of Lemma 2. We show that this implies that within any $n/16$ time steps each cluster is queried at least once with high probability. The main challenge lies in refreshing the knowledge about the clusters. The data structure internally maintains a clustering $D_1, \ldots, D_k$. However, when it queries some cluster $C$, it is not clear which of $D_1, \ldots, D_k$ does $C$ correspond to. To deal with that we show that the number of changes in each cluster within $n/16$ time steps is so low (again, with high probability), that there is a single cluster $D \in \{D_1, \ldots, D_k\}$, for which $|D \cap C| > |C|/2$.

The data structure of Lemma 3 can be used to simulate queries for cluster in the following way. Assume we want to discover the contents of cluster $i$. First, we use the data structure to get a node $v$, such that $\mathbf{Pr}(v \in C_i^t) \geq 1/2$. Then, we can use algorithm of Lemma 2 to get the entire cluster $C'$ of node $v$. Finally, we may use the data structure again to verify whether $C'$ is indeed $C_i^t$. This is the case with probability more at least $1/2$.

Moreover, the data structure allows us to assume that the algorithms are initially only given the number of nodes $n$ and the values of $p$ and $q$, because the data structure can provide to the algorithms both the number of clusters $k$ and their (approximate) probabilities.

## 4.2 Clustering Algorithms

Using the results of Section 4.1, we may now assume that algorithms may query the clusters directly. This allows us to give a simple clustering algorithm. The algorithm first computes a probability distribution $\rho_1, \ldots, \rho_k$ on the clusters, which is a function of the cluster probability distribution $\alpha_1, \ldots, \alpha_k$. Although the cluster probability distribution is not a part of the input data, we may use an approximate distribution $\alpha_1', \ldots, \alpha_k'$ given by the data structure of Lemma 3—this increases the error of the algorithm only by a constant factor. In each step the algorithm picks a cluster independently at random from the distribution $\rho_1, \ldots, \rho_k$ and queries it.

In order to determine the probability distribution $\rho_1, \ldots, \rho_k$, we express the upper bound on the error in terms of this distribution and then find the sequence $\rho_1, \ldots, \rho_k$ that minimizes this error.

**Theorem 4.** *Suppose that Assumption 1 holds. Then there exists an algorithm for clustering evolving graphs that issues $O(\log n/p)$ queries per step and that for each time step $t = \Omega(n)$ reports a clustering of expected error $O\left(\left(\sum_{i=1}^{k} \sqrt{\alpha_i}\right)^2\right)$. Furthermore if $p = 1$ and $q = 0$, it issues only $O(1)$ queries per step.*

The clusterings given by this algorithm already have low error, but still we are able to give a better result. Whenever the algorithm of Theorem 4 queries some cluster $C_i$, it finds the correct cluster assignment for all nodes that have been reassigned to $C_i$ since it has last been queried. These nodes are immediately assigned to the right cluster. However, by querying $C_i$ the algorithm also discovers which nodes have been recently reassigned *from* $C_i$ (they used to be in $C_i$ when it was last queried, but are not there now). Our improved algorithm maintains a *queue* of such nodes and in each step removes two nodes from this queue and locates them. In order to locate a single node $v$, we first discover its cluster $C(v)$ (using algorithm of Lemma 2) and then use the data structure of Lemma 3 to find the cluster number of $C(v)$. Once we do that, we can assign $v$ to the right cluster immediately. This results in a better bound on the error.

**Theorem 5.** *Assume that $\alpha_1 \geq \ldots \geq \alpha_k$. Suppose that Assumption 1 holds. Then there exists an algorithm for clustering evolving graphs that issues $O(\log n/p)$ queries per step and that for each time step $t = \Omega(n \log n)$ reports a clustering of expected error $O\left(\left(\sum_{1 \leq i \leq k} \sqrt{\alpha_i \sum_{i < j \leq k} \alpha_j}\right)^2\right)$. Furthermore if $p = 1$ and $q = 0$, it issues only $O(1)$ queries per step.*

Note that the assumption that $\alpha_1 \geq \ldots \geq \alpha_k$ is not needed for the theorem to be true. However, this particular ordering minimizes the value of the bound in the theorem statement.

Let us compare the upper bounds of Theorems 4 and 5. For uniform distributions, where $\alpha_1 = \ldots = \alpha_k = 1/k$, both analyses give an upper bound of $O(k)$, which means that on average only a constant number of nodes per cluster contribute to the error. Now consider a distribution, where $k = \Theta(\sqrt{n/\log n})$ and $a_i = \Theta(1/i^2)$ for $1 \leq i \leq k$. The error of the first algorithm is $O(\log^2 n)$, whereas the second one has only $O(1)$ expected error. Furthermore in some cases, the difference can be even bigger. Namely, let us define the distribution as follows. Let $k = \lfloor (n/\log n)^{2/3} \rfloor + 2$ and $\epsilon = ((\log n)/n)^{1/3}$. We set $a_1 = a_2 = (1 - \epsilon)/2$ and $a_i = \epsilon/(k - 2)$ for $3 \leq i \leq k$. Then, the error of the first algorithm is $O\left(\left(\frac{n}{\log n}\right)^{1/3}\right)$, but for the second it is still $O(1)$.

### 4.3 Lower Bound

Finally, we provide a lower bound for the problem of detecting clusters in evolving stochastic block model. In particular, it implies that in the case when $p = 1$ and $q = 0$ the algorithm of Theorem 5 is optimal (up to constant factor).

**Theorem 6.** *Every algorithm that issues one query per time step for detecting clusters in the evolving stochastic block model and runs for $n/\log n$ steps has average expected error*

$$\Omega\left(\left(\sum_{i=1}^{k} \sqrt{\alpha_i \sum_{j=i+1}^{k} \alpha_j}\right)^2\right)$$

We note here that Theorem 6 can be extended also to algorithms that are allowed $\beta$ queries by losing a multiplicative factor $1/\beta$. The proof is based on the observation that if the algorithm has not queried some clusters long enough, it is unaware of the nodes that have been reassigned between them. In particular, if a node $v$ moves from $C_i$ to $C_j$ at time $t$ and the algorithm does not query one of the two clusters after time $t$ it has small chances of guessing the cluster of $v$. Some nontrivial analysis is needed to show that a sufficiently large number of such nodes exist, regardless of the choices of the algorithm.

## 5 Experiments

In this section we compare our optimal algorithm with some benchmarks and show experimentally its effectiveness. More precisely, we compare three different strategies to select the node to explore in each step of our algorithm:

- the optimal algorithm of Theorem 5,
- the strategy that probes a random node,
- the strategy that selects first a random cluster and then probes a random node in the cluster.

To compare these three probing strategies we construct a synthetic instance of our model as follows. We build a graph with 10000 nodes with communities of expected size between 50 and 250. The number of communities with expected size $\ell$ is proportional to $\ell^{-c}$ for $c = 0, 1, 2, 3$. So the distribution of communities' size follows a power-law distribution with parameter $c \in \{0, 1, 2, 3\}$. To generate random communities in our experiment we use $p = 0.5$ and $q = 0.001$.

Note that in our experiments the number of communities depends on the various parameters. For simplicity in the remaining of the section we use $k$ to denote the number of communities in a specific experiment instance.

In the first step of the experiment we generate a random graph with the parameters described above. Then the random evolution starts and in each step a single node changes its cluster. In the first $10k$ evolution steps, we construct the data structure described in Lemma 3 by exploring the clusters of a single random node per step. Finally, we run the three different strategies for $25k$ additional steps in which we update the clusterings by exploring a single node in each step and by retrieving its cluster.

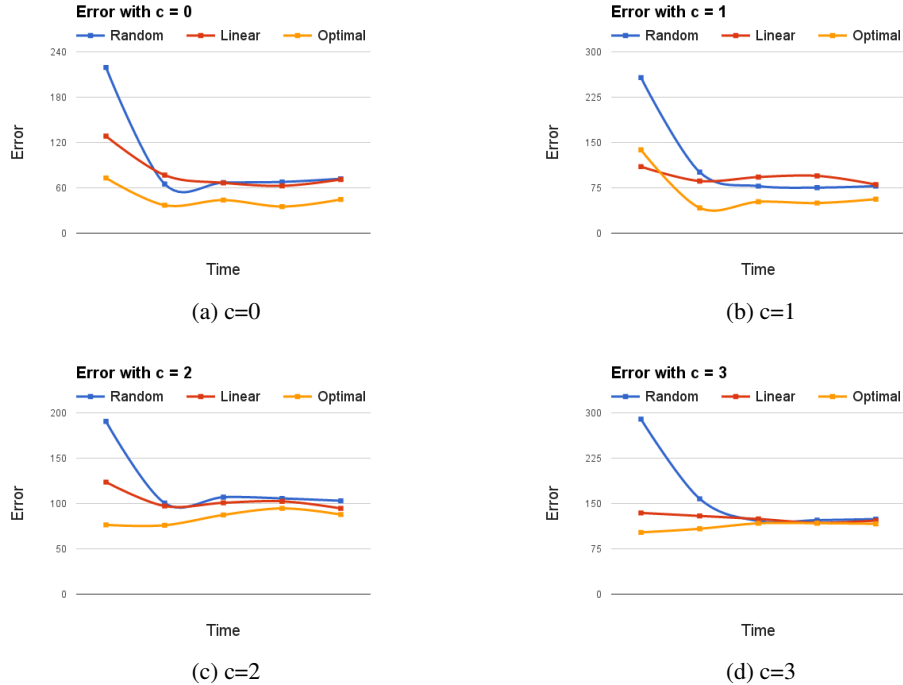

Figure 1: Comparing the performance of different algorithms on graphs with different community distributions.

At any point during the execution of the algorithm we compute the cluster of a node by exploring at most 30 of its neighbors.

In Figure 1 we show the experimental results for different values of $c \in \{0, 1, 2, 3\}$. We repeat all the experiments 5 times and we show the average value and the standard deviation. It is interesting to note that the optimal queue algorithm outperforms significantly all the other strategies. It is also interesting to note that the quality of the clustering worsens with time, this is probably because of the fact that after many steps the data structure become less reliable.

Finally, notice that as $c$ decreases and the communities' size distribution becomes less skewed, the performance of the 3 algorithms worsens and becomes closer to one another, as suggested by our theoretical analysis.

#### Acknowledgments

We would like to thank Marek Adamczyk for helping in some mathematical derivations. This work is partly supported by the EU FET project MULTIPLEX no. 317532 and the Google Focused Award on "Algorithms for Large-scale Data Analysis."

## Footnotes

[1]In our analysis we show that the assumption about the query model can be dropped at the cost of increasing the number of queries that we perform at each time step by a constant factor.

[2]Note that we can extend this definition to pairs of clusterings with different numbers of clusters just by adding empty clusters to the clustering with a smaller number of clusters.

[3]We define the term *with high probability* in Section 4.

[4]Usually, the constant $c$ can be made arbitrarily large, by tuning the constants of Assumption 1.

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
