[Reviews · NeurIPS 2016]

Reviewer 1

Summary

The paper presents a dynamic version of the well-known probabilistic block model. The task is to learn the community structure over time by probing information about the neighborhood of individual nodes. One desired objective is to keep the number of probes as small as possible. The authors present an evolution model under which they can bound the number of queries and the average expected error (measured in terms of "distance" between the computed clustering and the desired one).

Qualitative Assessment

While the dynamic and probing settings made in the paper are reasonable, the approach and the proposed model make a number of unrealistic assumptions. As a consequence it remains an open question how useful the whole apparatus can be in practical terms. The mutual independence of link connections of the basic block model is also carried out in the proposed approach and this assumption is clearly violated in practice. Furthermore, the re-assignment of nodes to clusters is carried out independently for each node, and independently of "its role" within a cluster. Probabilities p and q are assumed to be known - which is far from being true in practice. An open question that I would like to see discussed in the paper (also via experiments with real data) is how useful is the proposed model in practice? How well would it work when the underlying assumptions are violated? It also seems that, while nodes can migrate across community, they are required to "stay active" and maintain connections to other entities. In practice, not only nodes migrate and change associations to communities, but also become inactive. Can the proposed model account for this? What is the motivation for using the definition of distance between clusterings given on page 4? Why not using the commonly used mutual information to assess the quality of the estimated clustering?

Confidence in this Review

2-Confident (read it all; understood it all reasonably well)


Reviewer 2

Summary

The authors study in this paper clustering of a particular type of evolving graphs for which a query model is used to update the clustering through time. The authors give upper and lower bound on the number of queries needed to reach a certain accuracy in the recovery of the clustering structure.

Qualitative Assessment

The theoretical results proved in this paper are quite impressive, but I'm not sure they are really relevant in practice. The query model itself is realistic, but the evolving graph model is questionable (as is the specific version of stochastic block models used by the authors). It should be first noted that SBM does not assume a community structure with p > > q (using the authors' notation). In many application fields (e.g. biology), other structures are as important as communities (bipartite like structures, star structures, etc.) and all of them are particular cases of the SBM. The case p = 1 and q = 0, while nice on the theoretical results is irrelevant in practice. Then, the evolving graph model makes a very strong and unrealistic assumption: the connectivity structure (the probabilities q_{ij} and p_i) do not change through time. This means in practice that an actor can suddenly switch from one role to another one without retaining any of the property of its former role. This might be realistic in some specific contexts (maybe biology, for instance), but this is surely inadequate in many others (e.g., social networks). For those reasons, added to the fact that the paper comes with 20 pages of supplementary materials (without which the paper is difficult to follow), I don't think this paper is adapted to a conference and would be more suited for a journal publication.

Confidence in this Review

1-Less confident (might not have understood significant parts)


Reviewer 3

Summary

This work studied a model of clustering on evolving graphs, where the graphs themselves and clustering assignment for each node are changing over it. The model is based on the traditional stochastic block model, where the connection between two nodes are characterized by their clustering assignments; Two nodes of the same community are linked with probability p; while the probability of a link between two nodes of different communities is q. Based on the evolving graphs, it assumes that each time we could only perform limited number of calls to query nodes and their neighbors. The clustering algorithm is designed upon the stochastic block model and the limited graph probing. Finally, it also provides upper bounds and a lower bound of clustering errors for the problem of detecting clusters in evolving stochastic block model.

Qualitative Assessment

This work studied the problem of node clustering (community detection) on dynamic evolving graph. The problem itself is very interesting, and this work is very solid and provides interesting theoretical results. However, the major issues lie on the motivation of the setting, the limitation of the stochastic block model, and the structure of the paper. Details are discussed as follows: 1. It is not convinced that why we only perform limited number of calls to query nodes and their neighbors. Usually in reality, we admit the observed graphs have missing nodes and links, and develop graph algorithms over those imperfect settings with recovering hidden information. 2. In addition, this setting makes the problem look very similar to row sampling for low-rank graph matrix approximation, with extension to the dynamic setting or on-line setting. The ground truths graph matrices are G1, \cdots, Gt, we could only perform limited row sampling (the sampled rows corresponding to the neighborhood query to nodes) and low-rank approximation, to approximate G1, \cdots, Gt. The low-rank matrix exactly denotes the node clustering assignment. Therefore, those theoretical results for low-rank approximation with row sampling might be connected to the presented results. 3. The stochastic block model is very limited, especially in the case where p=1 and q=0. Is there any reason that not using mixed membership model or other general model? 4. The structure of the paper needs to be reorganized: 1) It would be appreciated if moving the algorithms from supplement to the main text, and then discuss the properties of desired data structure used in the algorithm to support efficient graph probing (Lemma 3). 2) It would be appreciated about discussing some intuitions of Lemma 1, 2 such as how much we could recover the entire network regarding to XX number of queries. 5. The experiments did not well support claims and theoretical results. It only compared the proposed querying strategy to random node query and node query proportional to cluster size. We expected to see: 1) How the clustering results change with number of queries? 2) Since the experiments are conducted on synthetic networks with ground truth clusters, compare the clustering results returned by the algorithm to the ground truth Compare the theoretical bounds to the empirical results.

Confidence in this Review

2-Confident (read it all; understood it all reasonably well)


Reviewer 4

Summary

The authors provide an algorithm for performing clustering on evolving graphs. The authors assume a stochastic block model for clustering the graphs and their model of evolution is that one node changes its clusters and all neighbors with a given (fixed) probability distribution. Their query model is local: a query will reveal a node and all its neighbors. Their algorithm makes a small number ~ O(log n / p) of queries each round and, after a initializing for O(n log n) steps, is able to keep its view of the clusters and associated probabilities accurate for O(n^2) steps (with low expected error in most interesting settings). The authors also present a proof of optimality of their algorithm in their settings and perform an experiment on synthetic data to demonstrate the working of their algorithm.

Qualitative Assessment

I believe that the algorithm presented in the paper is novel and would lead to significant follow-up work. The analysis of the setting is technically challenging and is handled is in-depth. The results are also conveyed well. For instance, the authors identifying two different parts of the problem (viz. discovering cluster of a given node and creating/maintaining a clustering), both of which allow for different algorithms to be placed in them (one which depends on whether p,q = 1,0 or not and the other which offers an improvement in the error guarantees). Their analysis is able to covers all combinations of these methods in a clear way. During the work up to their main result, the authors first present a simple each to understand clustering algorithm (with a complete analysis) and then improve it, highlighting the improvements. The motivations for the changes are also presented well (in the supplementary material). However, the writing of the paper is a little confusing at places, which is probably an artifact of restricted space. For example, it is hard to see how one of the observations they mention in the introduction (viz. strategy of choosing clusters to query) follows without knowing their model in more detail. Similarly, claim in line 168 is not justified unless we know how to fix K first. Also, the paper lacks in-depth experiments and discussion of results. Their algorithm seems to performs well on synthetic data, but a demonstration on a real world setting would be more convincing. The graphs have confusing legend and no (visible) error bars. They do not compare the performance of their algorithm against their own simpler benchmark algorithm, nor do they explain why the error in classification falls sharply after the initialization in some cases (I suspect it is because the enqueuing of mis-classified type II nodes starts happening, as per Algorithm 5)? Minor typos: - line 710, it should be "we denote the second index ..." instead of "we denote the first index ..." - line 282: "changes iof guessing" - line 181: "time steps" -> "time step" - line 137: "independent with" -> "independent of" - line 93: "asymptoticall"

Confidence in this Review

1-Less confident (might not have understood significant parts)


Reviewer 5

Summary

The paper considers community detection in an evolving graph setting where the nodes and edges of the graph are changing over time but are not observed. Instead, observations are gathered using a limited number of probes or queries of the neighbors of individual nodes. The authors consider a simple model for evolving the graph by augmenting a stochastic block model with community changes and provide theoretical results showing that their proposed probing strategy is asymptotically optimal in a special case and also performs well in simulations.

Qualitative Assessment

Strengths: + Among the first papers that I know of to provide theoretical results for estimation in dynamic graph models. This is an important first step that I hope will motivate further research in this area. + Considers a partial observation/active sensing setting that has not been explored much in the community detection literature. + The paper is written in a very accessible manner, and a lot of theory has been condensed and explained in a fluid and coherent way. Weaknesses: - The theoretical results don't have immediate practical implications, although this is certainly understandable given the novelty of the work. As someone who is more of an applied researcher who occasionally dabbles in theory, it would be ideal to see more take-away points for practitioners. The main take-away point that I observed is to query a cluster proportionally to the square root of its size, but it's unclear if this is a novel finding in this paper. - The proposed model produces only 1 node changing cluster per time step on average because the reassignment probability is 1/n. This allows for only very slow dynamics. Furthermore, the proposed evolution model is very simplistic in that no other edges are changed aside from edges with the (on average) 1 node changing cluster. - Motivation by the rate limits of social media APIs is a bit weak. The motivation would suggest that it examines the error given constraints on the number of queries. The paper actually examines the number of probes/queries necessary to achieve a near-optimal error, which is a related problem but not necessarily applicable to the social media API motivation. The resource-constrained sampling motivation is more general and a better fit to the problem actually considered in this paper, in my opinion. Suggestions: Please comment on optimality in the general case. From the discussion in the last paragraph in Section 4.3, it appears that the proposed queue algorithm would is a multiplicative factor of 1/beta from optimality. Is this indeed the case? Why not also show experiment results for just using the algorithm of Theorem 4 in addition to the random baselines? This would allow the reader to see how much practical benefit the queue algorithm provides. Line 308: You state that you show the average and standard deviation, but standard deviation is not visible in Figure 1. Are error bars present but just too small to be visible? If so, state that it is the case. Line 93: "asymptoticall" -> "asymptotically" Line 109: "the some relevant features" -> Remove "the" or "some" Line 182: "queries per steps" -> "queries per step" Line 196-197: "every neighbor of neighbor of v" -> "neighbor of" repeated Line 263: Reference to Appendix in supplementary material shows ?? Line 269: In the equation for \epsilon, perhaps it would help to put parentheses around log n, i.e. (log n)/n rather than log n/n. Line 276: "issues query" -> I believe this should be "issues 1 query" Line 278: "loosing" -> "losing" I have read the author rebuttal and other reviews and have decided not to change my scores.

Confidence in this Review

2-Confident (read it all; understood it all reasonably well)